# Willingness to Pay for Public Benefit Functions of Daecheong Dam Operation: Moderating Effects of Climate Change Perceptions

**Heekyun Oh**, **Seongjun Yun** and **Heechan Lee** *

College of Hospitality and Tourism Management, Sejong University, 98 Gunja-Dong, Gwangjin-Gu, Seoul 143-747, Korea; heekyunoh@sejong.ac.kr (H.O.); jun_yun1126@naver.com (S.Y.)
* Correspondence: leeheech@sejong.ac.kr

**Abstract:** This study estimates the economic value of the Daecheong Dam for the public function of responding to climate change. It examines the moderating effect of climate change perceptions on value estimates by applying choice experiments (CE). The study specifies three dam function attributes—drought management (DM), flood control (FC), and water quality monitoring (WM)—subdivided into three levels to improve the existing conditions. Survey data from 603 households living in Daejeon, Chungbuk, and Chungnam have been collected to perform the CE. Subsequently, two clusters—high-involvement and low-involvement groups—have been extracted, based on the climate change perception index. The main results of comparing the marginal willingness-to-pay between the two clusters are as follows. The attributes and price variable significantly affected the choice probability to benefit from improvements in the rational signs of the coefficients. This does not violate the independence of the irrelevant alternatives assumption. The improvement values of high-involvement and low-involvement groups are estimated as KRW 21,570 and KRW 14,572 a year per household, respectively. Both show the same value intensities in the order of WM, DM, and FC.

**Keywords:** choice experiments; climate change; Daecheong Dam; public functions; willingness to pay (WTP); non-market values





## 1. Introduction

Global climate change has intensified precipitation irregularity, lake and river surface decline, and water quality deterioration [1–3], which in turn has hampered water management efficiency [4,5]. In South Korea, drought and flood damage keeps recurring [6–8]. Hence, countermeasures against climate change are being promoted. One of the highlighted issues is dam operation [9,10]. Recently, since water supply and demand management in response to climate change has become a national problem, a variety of measures have been proposed for effective water resource management. This includes restructuring the main role of hydroelectric dams to supply water during drought, and flood defenses [11]. Efficient dam operation plans are urgently required to manage drought, flood stress, and water quality. Multi-purpose dams benefit local people, directly and indirectly, by providing domestic and industrial water, electricity generation, and eco-tourism as well as drought relief and flood prevention [12,13]. Furthermore, the fact that the reservoir water condition is highly relevant to drinking water quality and recreational value for local people has added significance [14,15].

The point of interest here is that water supply and distribution should be government controlled, since the benefits of using dams are characterized by public goods more than private goods [16]. Therefore, it has become a major concern to confirm the input cost validity (The feasibility of the dam project is determined by a cost and benefit economic analysis, and the result of comparing these two figures affects investment decisions [17].) when implementing dam operational improvement projects for public use. Such procedural

justification can be ensured in case the benefit exceeds the cost [17,18]. At this point, as the operational benefits (including drought prevention, flood protection, and water quality management) [19–21] are services for unspecified individuals, public interest valuations are eventually considered for judging government project performance. First, given that time or cost constraints are unavoidable, examining core factors of value inducement and identifying influencer priorities might be regarded as important to foster business efficiency.

Several relevant studies have emphasized the importance of identifying climate change impacts on water management. Vital research problems about the economic value of drought stress alleviation, flood risk management, and water quality improvement have been globally discussed, thus, contributing to the awareness of the economic value of the public benefits provided by dam functions. However, the results of these studies did not examine the direct value. Furthermore, it is difficult to immediately compare results from different analysis environments due to different spaces and timeslots.

Therefore, this study primarily investigates the economic value of the role of dams in coping with climate change and benefiting the public through drought management (DM), flood control (FC), and water quality monitoring (WM). Thus, it ultimately provides foundational information through which to highlight the public benefits of establishing water-resource countermeasures against climate change. Considering the severe damage caused by floods, droughts, and water pollution by South Korea's changing climate, dealing with the three functions is desirable. In addition, the study investigates the moderating effect of climate change risk perceptions on the economic value of each public function; the perceived public value can vary significantly according to the climate change awareness level. Confirming whether there are discriminative values is considered necessary for highly acceptable policy drives. Some studies (e.g., [22–24]) have shown that climate-change awareness affects the acceptance of dam operation policies. Thus, generalizing the results of the study without additional verification regarding the subdivided value might distort the value judgment.

This study investigates the benefits of the Daecheong Dam. The Daecheong Dam—completed in 1980 as a multi-purpose dam—is 72 meters tall and is 495 meters wide. Its catchment area is 4,134 square kilometers with a capacity of about 1.49 billion cubic meters. The reservoir, formed by dam administrators, is located within Chungcheongnam-do and Chungcheongbuk-do. As a serious water-bloom phenomenon increased after the 2015 dry season, there was an emergency in water quality management regarding Lake Daecheong—the source of drinking water for the Chungcheong region. The waterworks authority exercised closer monitoring of harmful algal blooms at Daecheong reservoir [25,26], located in Daejeon metropolitan city, South Korea. The public role of the Daecheong Dam, directly and indirectly, include the benefits of water supply for agricultural, industrial, and residential use, as well as the supervision of water quality for drinking and recreation [27–29]. Drought, flood, and water pollution emphasize how crucial dam operation plans can be. Moreover, this study applied the choice experiment (CE) (The estimation of economic values for public services is carried out in diverse environmental fields (e.g., [30,31]), and CVM and CE have been regarded as typical valuation methods. Among the several econometric methods, the CE designed by Adamowicz et al. [32] has the advantage of subdividing the value of the estimated object into main attributes. Moreover, progressive values can be estimated by phases from the lowest to the highest level. The bundle of alternatives combined by each level of functionality is presented to respondents, after which the most preferred alternative (including a price level) is selected (this can be calculated as the values for each level). In particular, where the effects of water resources development plan vary, CE can be cost-effective by enhancing the feasibility of policy alternatives.) methodology for valuation, which has the advantage of separating the attributes that affect the value of a certain good by level and estimating the value of each level [33].

The study concluded that three dam functions are of high importance through several key pieces of evidence, and verifying them is crucial for South Korea. The role of dams has been specified through three major attributes: DM, FC, and WM. Furthermore, the

results of advanced studies (which suggest the perceived seriousness of climate change significantly affects policy support ratings) indicate that the higher the level of consideration in climate change, the greater the likelihood of advocating for the enhancement of dam functions [22–24]. Accordingly, it is expected that people's awareness of climate change may cause meaningful differences in the economic values of the dam's public functions. Hence, the study conducts empirical analyses to achieve specific objectives as follows:

1. Estimate the economic value of Daecheong Dam by the subdivided attributes (DM, FC, and WM).

2. Examine the moderating effect of climate change perceptions on the economic values of the dam's public functions.

This study contributes to the literature by estimating the WTP of Daecheong Dam's functions and how they differ depending on climate change awareness. Taking the two previously mentioned objects into consideration, the following research questions are proposed.

$Q_1$. Do the three attributes (DM, FC, WM) of Daecheong Dam have a significant impact on the increase in the utility of survey respondents?

$Q_2$. Does the MWTP for the public interest function of Daecheong Dam differ depending on the degree of awareness of climate change among survey respondents?

## 2. Materials and Methods

### 2.1. Study Area

The Daecheong Dam basin (36°28′33.3″ N 127°28′31.1″ E) is 2608 km$^2$, accounts for more than 1/4 of the total area of 9914 km$^2$ of the Geumgang River basin, and 10 administrative districts. It spans Daedeok-gu, Dong-gu, Yuseong-gu, Daejeon Metropolitan City; Cheongju-si, Boeun-gun, Okcheon-gun, Chungcheongbuk-do; Geumsan-gun, Chungcheongnam-do, Yeongdong-gun; Sangju-si, Gyeongsangbuk-do and Muju-gun, Jeollabuk-do [34] (see Figure 1). In the Daecheong Dam basin, there are 108,852 residents from across 43,140 households, and the population density was analyzed to be 163.08 people/km. The water supply rate in the Daecheong Dam basin was found to be about 86.0%, lower than Korea's total water supply rate, 96.5% (including village water supply, small water supply facility population) [35].

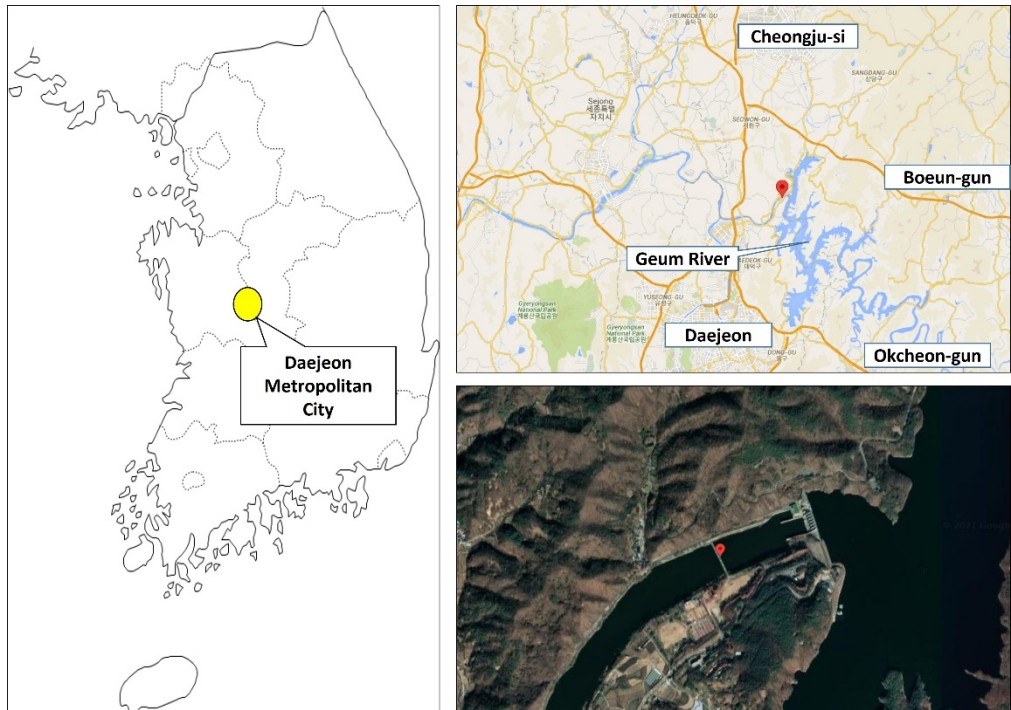

**Figure 1.** Location of the Daecheong Dam basin.

Construction for the Daecheong Dam began in March 1975, and it was completed in December 1980. It is a complex dam composed of a gravity-type concrete dam and a sand dam with a height of 72 m, a length of 495 m, a reservoir area of 72.8, and a volume of 1,234,000 m$^3$. There exists the main dam with a storage capacity of 1.49 billion m$^3$, and three auxiliary dams that prevent water in the reservoir from overflowing to other areas. In addition, there are hydroelectric power plants with a capacity of 90,000 kW and a water channel to supply water to some areas of the Chungcheong region.

*2.2. Literature Review*

There is a large body of scientific data linking climate change to hydrologic changes such as in precipitation, streamflow and evapotranspiration. Climate change's impact on the hydrologic cycle poses a severe threat to Korea, an area threatened by periodic floods and droughts. Climate change-induced increases in streamflow during the monsoon (a period of significant rainfall, typically May–September) have the potential to exacerbate flood damage, whereas increases in evaporative losses (due to warmer temperatures) during the dry period can exacerbate water scarcity in some areas [36].

Among the measures used to manage water resources in response to climate change, the operation of dams and the improvement of their functions has emerged as a major subject of interest. In the case of multipurpose dams, the importance of dam operation plans for flood control, drought management, and environmental functions is increasing as they are directly or indirectly related to the benefits of water for residents, such as water supply, hydroelectric power generation, and water quality improvement [37,38]. The public interest value of water resources by the dam function makes it difficult to clearly measure benefits, and the absence of a market has acted as a challenge in efficient resource distribution, making government intervention inevitable in water supply and distribution. For this reason, evaluating and proving the validity of non-market value for dam function and water resource use is considered a task that must be preceded in the process of controlling and managing it [39].

Because it is concerned with modeling options ranging over a variety of attributes rather than estimating WTP for a single option, the CE technique presents a potential chance to quantify the economic values of diverse environmental consequences induced by big dam development. The CE methodology, similar to the CV method's referendum model, has its theoretical grounding in the random utility model, which is compatible with economic theory [40–42].

The rationale for estimating the values of dam functions regarding DM, FC, and WM is sourced from various prior studies. First, value estimation studies resulting from drought mitigation are typically conducted in terms of drought relief for watershed protection [43], the willingness to pay (WTP) to avoid drought-water constraints for households and businesses [44], premium payments for agricultural insurance [45], and the value of avoiding drought water-usage restrictions [46].

Furthermore, in studies on flood-risk reduction values, empirical tests on nationwide flood control measures [47], flood risk reduction [48], flood insurance premiums for rural households [49], and the economic value and determinants of flood defenses [50] were explored. Moreover, regarding water quality values, various studies on the value of secure and reliable drinking water [51], the amount of payments to improve in-home water services [52], the value of water quality improvement and determinants that affect the value [53], and the quality improvement value of tap water for urban residents [54] have been carried out. In most of the previous studies mentioned, the contingent valuation method (CVM) and CE were used for measurement. It was also noted that individual characteristics such as gender, age, income level, education level, residential environment, government trust, and perceptions about disasters affected the value determination [45,47–51,53,54].

While a variety of economic valuation cases have been globally executed, it has been confirmed that there are few intermittent studies in South Korea. The precedent studies

relevant to the three roles are as follows. Hwang et al. [55] estimated WTP to improve the future status of Korean water scarcity by households using CVM. Thus, Busan residents in Korea perceived water shortage, and about 70% of them were willing to pay. The average payment amount per household was about KRW 3572 (USD 4) per household per month. Choi and Lee [56] calculated home buyer contributions to flood prevention construction through the hedonic price method. According to the results, the buyer's WTP for a 1% reduction in rainfall intensity was KRW 62,101 per square meter, and the WTP for a 1% reduction in annual rainfall was KRW 36,533 per square meter.

Furthermore, Lee et al. [57] used CVM to evaluate the WTP for a future water shortage project in Korea, resulting in about 320 million dollars. They, however, concluded that the project cost was greater than the national utility. Kwak et al. [58] valued WTP for tap-water quality improvement in Busan, Korea, through CVM; the average amount per household was KRW 2124 per month. In addition, Um et al. [59] applied the averting behavior method to estimate WTP to reduce the negative perceptions caused by the discrepancy between the objective pollution level and perceived level. The results highlighted that perceived risk is more effective than objective risk, and the USD range of WTP were [0.07; 1.70] to [4.2; 6.1].

Moreover, these three functions act as major factors of dam operations according to an expert opinion survey that prioritizes the core properties for adapting to climate change. Furthermore, there is much emphasis on paying constant attention to comprehend the managerial importance of these factors [60]. So far, it is clear that CVM, CE, and the hedonic price method were frequently employed as value-estimation methods. CVM, which measured only the single value of the goods, was used most. In addition, the spatial and temporal features of the study site and the demographic characteristics of the study subjects had a significant effect on the estimation results.

### 2.3. Setting Attributes and Levels

Concerning the attributes from the previous studies, a content validity examination was further conducted. Thus, those three functions were selected as the final attributes based on carefully reviewed outcomes by experts (professors and senior researchers on environmentology, hydrology, and mineral economics). Focus group interviews (with ten regular people cognizant of Daecheong Dam) were, then, employed to determine specific levels of the attributes. In this respect, interviewees described the image associated with Daecheong Dam's climate change role. Accordingly, functional levels expressible in are cognizable manner were established. Information on techniques relevant to drought mitigation (such as sedimentation reduction and emergency drainage design), flood reduction (such as spillway design and dam raise), and water quality monitoring (such as the installation of devices for reducing non-point sources and sewage treatment facility expansion) was given to the interviewees in advance.

At the end of the discussion, the decision was that it is too restrictive to manifest the diffusion of specific technologies at a certain level. Thus, it was desirable to describe the attribute levels as complementing overall current technologies and creating new crafts beyond the present structure. Subsequently, each attribute level is classified into three phases: low-level (to maintain the status quo), medium-level (to complement existing technologies), and high-level (to develop new technologies along with the complementation). These demonstrate utilities calculated as per the increase in the improvement levels. In addition, a preliminary test for 30 respondents regarding WTP was conducted using open-ended questions to determine appropriate bid levels along with a realistic payment vehicle. Thus, via the focus group interviews and the reviewed attributes [61–63], it was determined that three asking prices of KRW 5000, KRW 10,000, and KRW 20,000 within the range of 15% to 82% of the response distribution [64] should be the annual financial support. The levels are shown in Table 1.

**Table 1.** Description of attribute levels.

| Attributes | Improvement Levels | | |
| --- | --- | --- | --- |
| | **Low** | **Medium** | **High** |
| Drought Management | (Status quo) Maintaining current techniques to prevent drought disaster | (Partial improvement) Complementing existing techniques to prevent drought disaster | (Substantial improvement) Complementing existing techniques and developing new techniques to prevent drought disaster |
| Flood Control | (Status quo) Maintaining current techniques to prevent flood disaster | (Partial improvement) Complementing existing techniques to prevent flood disaster | (Substantial improvement) Complementing existing techniques and developing new techniques to prevent flood disaster |
| Water quality Monitoring | (Status quo) Maintaining current purification techniques to prevent water pollution | (Partial improvement) Complementing existing techniques to prevent water pollution | (Substantial improvement) Complementing existing techniques and developing new techniques to prevent water pollution |

*2.4. Development of a Measurement Instrument*

The survey questionnaire was composed of demographic items (gender, age, marriage status, education, household income, and resident area), climate change perception index, and CE elements. Choice sets are first structured based on the derived attributes and levels to develop the measurement tool for CE. The procedures are as follows. Since the three attributes of the dam's public benefit function and the annual financial support, respectively include three levels, a total of 81 alternatives exist (3 raised to the 4th power). The study employed a more efficient experimental design using the SAS orthogonal design program because it is an unrealistic field survey that requires responses to all of the alternatives. Thirty-four optimal profiles were extracted, and 18 choice sets were derived from each choice profile involving two optional alternatives along with a reference alternative. Furthermore, the results are confirmed to be statistically significant due to superiority in terms of efficiency and error (D-efficiency = 2.08; D-error = 0.48) [65]. Presenting a set of 18 optional alternatives to one respondent may increase non-sampling errors. In this study, after dividing the entire survey questionnaires into Type A/B/C, six sets of optional alternatives were assigned to each type.

However, if one respondent evaluates all 18 choice sets at once, the response validity might be impacted. Thus, the sample was divided into three blocks to enhance the response validity. Each of the three questionnaire types contained six choice sets. Figure 2 below shows one of the 18 choice sets. The respondents evaluated the choice sets composed of each level of the dam's public benefit function and the annual financial support. Then, they selected the most preferred alternative among two options for further improvement along with one "no-choice" option. Here, the level of each attribute in the "no-choice" option is low (i.e., status quo), and the annual financial support is designated as KRW 0. Respondents choose the most preferred alternative among the three options after reading the contents of the current technology level described in the questionnaire introduction.

Prior to the analysis, Option 3. "Choosing neither option" (see Figure 2) in the questionnaire indicates not selecting any of the two improvement alternatives, implying that the current condition would be maintained. Therefore, the willingness to pay financial support is calculated as KRW 0, but the water expense is still maintained.

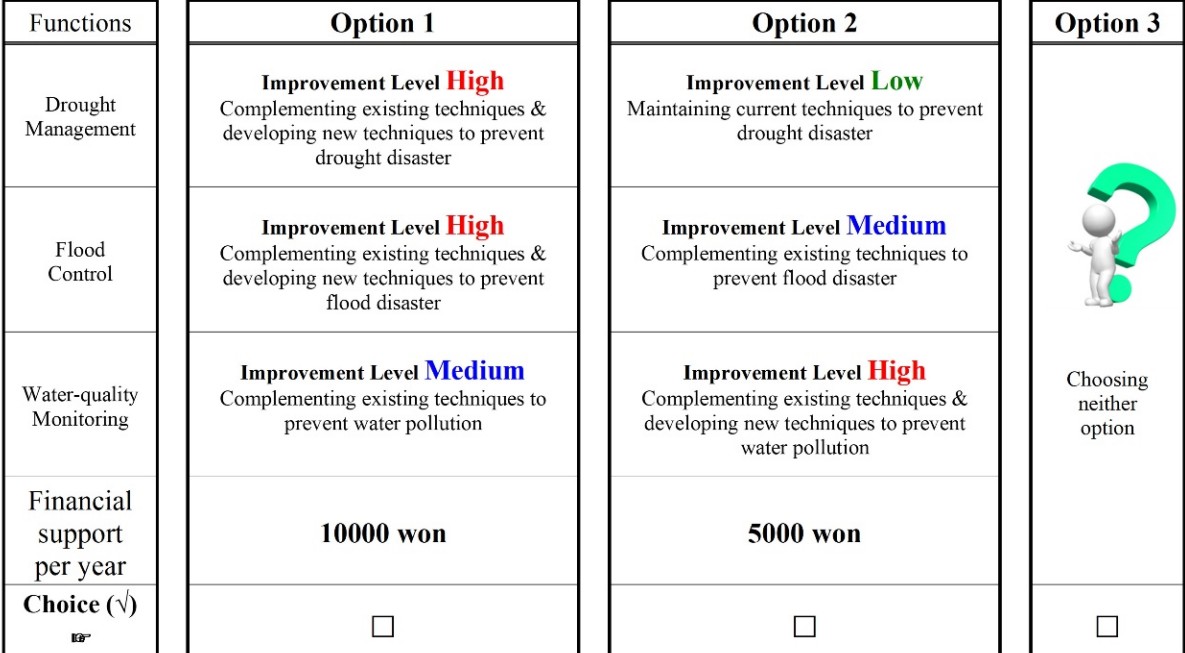

**Figure 2.** An example of choice set.

### 2.5. Sample Collection

The study population comprised adults aged 20 and older who, directly and indirectly, benefit from various water supply and hydroelectric power generation of Daecheong Dam. For this reason, residents who were aware of Daecheong Dam and living in Daejeon City or its surrounding areas such as Chungbuk and Chungnam were selected as participants. The samples were selected according to gender and age-group properly represented the population (purposive quota sampling). Before the main survey, we conducted a pre-test to check whether the content, arrangement, and phrasing of the items were clear. We completed the final questionnaire by correcting and complementing the questions. A total of 630 questionnaires (210 copies for each type) were distributed under the household unit of analysis by direct face-to-face street-intercept interviews around Geum River, LOHAS Park, near Daecheong Dam. After screening, 603 valid questionnaires were employed for data analyses.

### 2.6. Analytical Method

The analytical model of CE is based on the indirect utility function theoretically implied in economics. The function $U_{ij}$ in Equation (1) indicates the indirect utility of any individual $i$ (= 1, . . . , n), which can be obtained from an alternative $j$ (= 1, . . . , J) among a choice set $C_i$.

$$U_{ij} = V_{ij}\,(Z_{ij},\,S_i) + e_{ij} \tag{1}$$

Here, $V_{ij}$ accounts for the attribute functions of the alternative ($Z_{ij}$) and the individual characteristics ($S_i$) of the respondent as the observable elements. In addition, $e_{ij}$ means unobservable errors, which are relevant to the theoretical foundation for composing the likelihood function.

In the CE analysis, the discrete choice model is applied. If the $j$th alternative of the choice set $C_i$ chosen by the respondent $i$ generates a greater utility than another alternative $k$ [$U_{ij} > U_{ik}$ ($k \in C_i$, $k \neq j$)], it is logically clear from the above that the alternative $j$ should be chosen. Thus, in Equation (2), the probability of respondent $i$ choosing alternative $j$ can be written as:

$$P(j\,|\,C_i) = Pr\,(V_{ij} + e_{ij} > V_{ik} + e_{ik}) = Pr\,(V_{ij} - V_{ik} > e_{ik} - e_{ij}) \tag{2}$$

In the case of estimating the multi-nominal logit model described in Equation (2), if the assumption about the error term independence is satisfied (per the Type I extreme value distribution), then the probability of the respondent *i* selecting alternative *j* is given by Equation (3).

$$P_i(j|C_i) \ = \ \frac{exp(V_{ij})}{\sum_{k \in C_i} exp(V_{ik})} \tag{3}$$

The multi-nominal responses derived from the CE questionnaire represent the outcomes where the individuals pursue utility maximization. This is analyzed through the likelihood function in Equation (4).

$$lnL = \sum_{i=1}^{n} \sum_{j=1}^{J} \left\{ Y_{ij} \cdot ln[Pr_i \, (j|C)] \right\} \tag{4}$$

In this case, the respondent may or may not select alternative *j*, where the variable $Y_{ij} = 1$ indicates that the *i*th respondent has chosen the alternative *j*. Here, $1(\cdot)$ denotes the indicator function, and "1" is assigned in $1(\cdot)$ when the *j*th alternative is selected; otherwise, 0 is granted. Hence, the parameters can be calculated by applying the method of maximum likelihood estimation to the log-likelihood function of Equation (4) [66].

The indirect utility function $V_{ij}$ of this study can be described as the linear function of observable attribute vectors: an alternative specific constant (*ASC*), medium level ($DM_{Mid}$) and high level ($DM_{High}$) for the DM function improvement, medium level ($FC_{Mid}$) and high level ($FC_{High}$) for the FC function improvement, medium level ($WM_{Mid}$) and high level ($WM_{High}$) for the WM function improvement, and financial support (*Bid*) as shown in Equation (5). $\beta$ is an estimated parameter that affects the utility.

$$V_{ij} = ASC + \beta_1 DM_{Mid,ij} + \beta_2 DM_{High,ij} + \beta_3 FC_{Mid,ij} + \beta_4 FC_{High,ij} + \beta_5 WM_{Mid,ij} + \beta_6 WM_{High,ij} + \beta_7 Bid_{ij} \tag{5}$$

Moreover, an extended model into which demographic variables are additionally inserted is estimated for detailed examinations. The model is structured as in Equation (6):

$$V_{ij} \ = \ ASC \ + \ \beta_1 DM_{Mid,ij} \ + \ \beta_2 DM_{High,ij} \ + \ \beta_3 FC_{Mid,ij} \\ + \ \beta_4 FC_{High,ij} \ + \ \beta_5 WM_{Mid,ij} \ + \ \beta_6 WM_{High,ij} \ + \ \beta_7 Bid_{ij} \ + \ \sum_{s=1}^{S} \gamma_s K_{si} \tag{6}$$

where $K_{si}$ is the vector representing the individual characteristics of the *i*th respondent, *s* $(= 1, \ldots, S)$ is the demographic variable, and $\phi$ is an estimate of the interaction variables.

Thus, the marginal willingness-to-pay (MWTP) for the attributes can be estimated by Equations (5) and (6), which demonstrate the marginal rate of substitution (The marginal rate of substitution can be defined as the quantity of one good to be discarded to obtain another [67], that is, respondents have to pay more for a higher level of improvement.) between each level of the attributes and the price variable. Therefore, the MWTP, owing to the vector variation of each attribute, can be estimated as the coefficient ratio of the corresponding level to the price variable as shown in Equation (7).

$$MWTP_{DM_{Mid}} \ = \partial Bid/\partial DM_{Mid} \ = \ -\beta_1/\beta_7 \\ \vdots \\ MWTP_{WM_{High}} \ = \partial Bid/\partial DM_{Hign} \ = \ -\beta_6/\beta_7 \tag{7}$$

This study employed a climate change perceptions index, proposed by the Korea Energy Management Corporation [68], to measure the climate change level cognized by the respondents. An R-type explanatory factor analysis (EFA), based on principal components, corroborated the measurement item validity. In the factor extraction process, only items higher than eigen value 1.0 were factorized with a loading of more than 0.4. To measure the reliability of measurement tools, an internal consistency technique using Cronbach's Alpha Coefficient was applied. If the value of the Cronbach's alpha coefficient is 0.6 or more, the

reliability can be considered valid, and the entire items can be analyzed by synthesizing them on a single scale.

## 3. Results and Discussion

### 3.1. Demographic Profile of the Sample

The demographic characteristics of the sample are shown in Table 2. In this survey, a total of 18 choice sets are split into three questionnaire types including 6 sets. Thus, a $\chi^2$ test for the condition of homogeneity between respondents was employed. The results demonstrated the alternative hypothesis that the collected data was heterogeneous; it was rejected at the 5% significance level for gender ($p = 0.980$), age ($p = 0.950$), marital status ($p = 0.694$), education ($p = 0.061$), occupation ($p = 0.497$), residence area ($p = 0.994$), and income ($p = 0.051$). Hence, it confirmed that there was no statistical difference regarding key demographic variables between groups according to the questionnaire types.

**Table 2.** Demographic profiles.

| Categories | Type A | | Type B | | Type C | | $\chi^2$-Test $p$-Value |
|---|---|---|---|---|---|---|---|
| | Frequency | % | Frequency | % | Frequency | % | |
| Gender | | | | | | | |
| Male | 106 | 52.2 | 106 | 52.5 | 102 | 51.5 | 0.980 |
| Female | 97 | 47.8 | 96 | 47.5 | 96 | 48.5 | |
| Age | | | | | | | |
| 20–29 | 57 | 49.8 | 55 | 44.6 | 57 | 47.0 | |
| 30–39 | 60 | 50.2 | 63 | 55.4 | 60 | 53.0 | |
| 40–49 | 55 | 28.1 | 54 | 27.2 | 54 | 28.8 | 0.950 |
| 50–59 | 21 | 29.6 | 22 | 31.2 | 23 | 30.3 | |
| 60s or older | 10 | 27.1 | 8 | 26.7 | 4 | 27.3 | |
| Marital status | | | | | | | |
| Single | 101 | 10.3 | 90 | 10.9 | 93 | 11.6 | 0.694 |
| Married | 102 | 4.9 | 112 | 4.0 | 105 | 2.0 | |
| Education | | | | | | | |
| Middle school or less | 6 | 3.0 | 2 | 1.0 | 6 | 2.9 | |
| High school | 78 | 38.4 | 53 | 26.2 | 74 | 35.6 | 0.061 |
| College degree | 94 | 46.3 | 120 | 59.4 | 94 | 45.2 | |
| Postgraduate degree | 25 | 12.3 | 27 | 13.4 | 34 | 16.3 | |
| Occupation | | | | | | | |
| Profession | 16 | 55.2 | 26 | 56.9 | 19 | 57.1 | |
| Clerical work | 69 | 26.6 | 78 | 25.2 | 70 | 24.7 | |
| Production | 14 | 18.2 | 10 | 17.8 | 18 | 18.2 | |
| Service | 17 | 7.9 | 11 | 12.9 | 15 | 9.6 | |
| Civil servant | 5 | 34.0 | 5 | 38.6 | 3 | 35.4 | 0.497 |
| Teaching staff | 4 | 6.9 | 3 | 5.0 | 4 | 9.1 | |
| Self-ownership | 16 | 8.4 | 7 | 5.4 | 17 | 7.6 | |
| Student | 30 | 2.5 | 31 | 2.5 | 28 | 1.5 | |
| Unemployed | 15 | 2.0 | 12 | 1.5 | 5 | 2.0 | |
| Housewife | 17 | 7.9 | 19 | 3.5 | 19 | 8.6 | |
| Residence area | | | | | | | |
| Daejeon | 112 | 14.8 | 115 | 15.3 | 113 | 14.1 | |
| Chungbuk | 54 | 7.4 | 51 | 5.9 | 49 | 2.5 | 0.994 |
| Chungnam | 37 | 8.4 | 36 | 9.4 | 36 | 9.6 | |
| Monthly household income (unit: 10,000 won) | | | | | | | |
| 99 or less | 7 | 3.4 | 5 | 2.5 | 5 | 2.5 | |
| 100–199 | 29 | 14.3 | 19 | 9.4 | 16 | 8.1 | |
| 200–299 | 35 | 17.2 | 29 | 14.4 | 48 | 24.2 | |
| 300–399 | 38 | 18.7 | 42 | 20.8 | 29 | 14.6 | |
| 400–499 | 36 | 17.7 | 41 | 20.3 | 32 | 16.2 | 0.051 |
| 500–599 | 16 | 7.9 | 24 | 11.9 | 28 | 14.1 | |
| 600–699 | 14 | 6.9 | 6 | 3.0 | 16 | 8.1 | |
| 700–799 | 9 | 4.4 | 16 | 7.9 | 13 | 6.6 | |
| 800 or more | 19 | 9.4 | 20 | 9.9 | 11 | 5.6 | |
| Total | 203 | 100 | 202 | 100 | 198 | 100 | - |
| $n = 603$ | | | | | | | |

### 3.2. Estimating Conditional Logit Model

Table 3 shows the estimation results for the conditional logit model. Model I is a basic model to which the attributes of the public functions—DM, FC, WM, and financial support—are solely assigned. Model II is an extended model with additional demographic variables because individual characteristics need to be used as control variables based on previous research that found that socioeconomic factors may influence the value estimates [69]. Thus, after analyzing 3618 observed data in both models, the basic model showed acceptable results. LLF was −3515.53 ($p < 0.001$), and the Pseudo R-squared (The Pseudo R-squared statistic, which provides an auxiliary explanation for the model fit, is not high. It is, however, preferable to highlight the figure because it tends to be lower than usual regression analysis. For instance, according to Brau [70], the 0.11 level is acceptable), was approximately 11.0%. Moreover, the price variable (*Bid*) was negatively effective at the 1% significance level, which satisfies the theoretical direction of the coefficient. All levels of the attribute variables (The levels named 'Mid' and 'High' of the three attributes indicate 'the change from the low to medium level' and 'the change from the low level to high level', respectively. Thus, those variables were coded as (1, 0) and (0, 1), where the low level signifies the reference alternative (0, 0)) have direct impacts at the 1% significance level (except $FC_{Mid}$ significant at the 5% level), which implies that the higher the attribute level, the greater the probability of choosing the proposed options compared to the status quo. That is, the enhancement of each function for the dam can increase its utility for local people.

**Table 3.** Estimates of conditional logit models.

| Model | Model I | | | | Model II | | | |
|---|---|---|---|---|---|---|---|---|
| Variable | *Coef.* | *S.E.* | *t*-Ratio | | *Coef.* | *S.E.* | *t*-Ratio | |
| ASC | 0.151 | 0.082 | 1.83 | * | 0.790 | 0.316 | 2.50 | ** |
| DM_Mid | 0.454 | 0.065 | 7.02 | *** | 0.453 | 0.065 | 7.00 | *** |
| DM_High | 0.626 | 0.059 | 10.57 | *** | 0.627 | 0.059 | 10.57 | *** |
| FC_Mid | 0.273 | 0.059 | 4.59 | ** | 0.274 | 0.060 | 4.60 | *** |
| FlC_High | 0.391 | 0.060 | 6.46 | *** | 0.393 | 0.061 | 6.49 | *** |
| WM_Mid | 0.567 | 0.061 | 9.31 | *** | 0.568 | 0.061 | 9.32 | *** |
| WM_High | 0.818 | 0.060 | 13.58 | *** | 0.818 | 0.060 | 13.59 | *** |
| Bid | −0.938 | 0.043 | −21.58 | *** | −0.939 | 0.044 | −21.58 | *** |
| ASC*Gender | | | | | 0.023 | 0.076 | 0.30 | |
| ASC*Age | | | | | −0.010 | 0.004 | −2.50 | ** |
| ASC*Income | | | | | 0.047 | 0.019 | 2.46 | ** |
| ASC*Education | | | | | −0.025 | 0.017 | −1.47 | |
| ASC*Marital Status | | | | | −0.016 | 0.077 | −0.21 | |
| ASC*Occupation | | | | | −0.123 | 0.089 | −1.38 | |
| ASC*Residence Area | | | | | 0.050 | 0.077 | 0.65 | |
| *LLF* | −3515.53 | | | | −3506.31 | | | |
| Adj. Pseudo $R^2$ | 0.110 | | | | 0.113 | | | |
| No. of Obs. | 3618 | | | | 3618 | | | |
| | Alternative dropped | | $\chi^2$ (*df* =7) | | | *p*-value | | |
| IIA test | Option 1 | | 9.166 | | | 0.241 | | |
| | Option 2 | | 6.589 | | | 0.473 | | |
| | Option 3 | | 12.613 | | | 0.082 | | |

Note (1) Model I $\Rightarrow$ Model II: $\chi^2$ (0.05, 7) = 18.44 > 14.07; Note (2) ***, **, *: Significance at the 1%, 5%, 10% levels, respectively.

Moreover, seven interaction variables between the alternative specific constant (*ASC*) and the demographic variables (Age, Income per household (unit: million KRW), and Education (years of education) are the continuous variables. Gender (with female = 0, male = 1), Marital status (with single = 0, married = 1), Occupation (unemployed = 0, employed =1), and residential area (Daejeon = 0, Chungnam and Chumgbuk = 1) are

dummy variables) were computed to identify other influencers on the choice probability which cannot be examined by the underlying attributes. Consequently, an extended model fitness was achieved as *LLF* (−3506.31) and *Pseudo-R*$^2$ (0.113) improved compared to the basic model. The age ($t = -2.50$) and income ($t = 2.46$) variables had a negative and positive influence at the 1% significance level. This implies that lower age and higher income levels mean more choice possibilities for improvement alternatives. However, performing the *Hausman* tests for the independence of irrelevant alternatives meant all *p*-values rejected the null hypothesis that parameter estimates are heterogeneous at the 5% significance level, confirming the independence of irrelevant alternatives (IIA) assumption regarding the independence of error terms had been fulfilled.

### 3.3. Measuring Climate Change Perceptions and Segmenting Respondents

The analysis revealed that three factors—cause, countermeasures, and effect—were derived in terms of the level of understanding, as in the theoretical composition. Moreover, the levels of awareness and practice are each a single factor. Bartlett's test of sphericity and the Kaiser–Meyer–Olkin values were statistically significant at the 0.01% level (Appendix A). In addition, the reliability tests showed *Cronbach α* values for all EFA factors to be more than 0.77—the level of understanding about the cause ($α = 0.776$), countermeasures ($α = 0.795$), the effect ($α = 0.808$), and the level of awareness ($α = 0.817$)—except for the level of practice with an $α$ value of 0.667, albeit close to 0.7 [71].

Subsequently, a two-step clustering analysis utilizing five such factors was conducted to distinguish the climate change perception segments from the whole group (see Table 4). Two clusters were derived. Due to an independent-samples *t*-test to reveal the features of the two clusters, it was classified into segments of high levels (*H*) and low levels (*L*) regarding the five factors. The "*H*" and "*L*" clusters were, respectively named as "high involvement" and "low involvement."

**Table 4.** Clustering and *t*-test according to climate change awareness.

| Clusters<br>Factors | | Cluster 1:<br>High Involvement | Cluster 2:<br>Low Involvement | *t*-Ratio |
|---|---|---|---|---|
| | | Mean(S.D.) | Mean(S.D.) | |
| Level of understanding<br>of climate change: | Understanding the causes | 3.46(0.36) H | 2.94(0.43) L | 15.82 *** |
| | Understanding the measure | 3.50(0.38) H | 2.88(0.35) L | 19.97 *** |
| | Understanding the results | 2.74(0.56) H | 2.35(0.51) L | 8.74 *** |
| Level of awareness of the behavioral pattern | | 4.16(0.51) H | 3.42(0.56) L | 16.29 *** |
| Level of behavioral style | | 3.54(0.54) H | 2.90(0.50) L | 14.68 *** |

Note (1) Statistical mean difference: L < H. Note (2) ***: Significance at 1% level.

### 3.4. Estimating Implicit Prices by Cluster

As shown in Table 5, the coefficients for MWTP calculation of each group were estimated based on the extended model. The conditional logit models of the two groups were compared based on the likelihood ratio test [72,73] to clarify the moderating effect according to climate change perceptions. First, the likelihood ratio test between the two models showed that $χ^2$ was 65.62. This is larger than the threshold of 30.58 at the 1% significance level with 15 degrees of freedom, proving that the moderating effect of climate change perceptions was effective between the two groups.

Regarding the demographic variable interaction with ASC, age ($p < 0.05$), income ($p < 0.01$), occupation ($p < 0.10$), education ($p < 0.10$), and residence area ($p < 0.05$), variables were solely significant in the high-involvement group, while the effect of age ($p < 0.10$), education ($p < 0.10$), and residence area ($p < 0.05$) was marginally revealed in the low-involvement group. Thus, the statistical differences in the determinants between the models were verified. Moreover, concerning the main attributes, there were significant effects in both groups, and the directions of influence were also the same. However, since

there is a limit to the comparison of the variable impacts on the significance or the effect size, slope tests were performed to address the statistical differences of the effects [74].

**Table 5.** Comparison of conditional logit models between clusters.

| Model | High-Involvement Group | | | | Low-Involvement Group | | | | Coef. Comparison | |
|---|---|---|---|---|---|---|---|---|---|---|
| Variable | *Coef.* | *S.E.* | *t*-Ratio | | *Coef.* | *S.E.* | *t*-Ratio | | | |
| ASC | 0.873 | 0.523 | 1.67 | * | 0.981 | 0.407 | 2.41 | ** | - | |
| DM_Mid | 0.637 | 0.108 | 5.92 | *** | 0.361 | 0.082 | 4.42 | *** | 4.17 | ** |
| DM_High | 0.846 | 0.098 | 8.61 | *** | 0.499 | 0.075 | 6.64 | *** | 7.89 | *** |
| FC_Mid | 0.274 | 0.096 | 2.86 | *** | 0.280 | 0.076 | 3.67 | *** | 0.00 | |
| FC_High | 0.389 | 0.098 | 3.98 | *** | 0.394 | 0.078 | 5.08 | *** | 0.00 | |
| WM_Mid | 0.638 | 0.099 | 6.47 | *** | 0.522 | 0.078 | 6.70 | *** | 0.84 | |
| WM_High | 1.005 | 0.097 | 10.33 | *** | 0.702 | 0.077 | 9.08 | *** | 5.95 | ** |
| Bid | −0.986 | 0.072 | −13.78 | *** | −0.922 | 0.055 | −16.69 | *** | 0.49 | |
| ASC*Gender | 0.095 | 0.126 | 0.75 | | 0.051 | 0.098 | 0.51 | | 0.08 | |
| ASC*Age | −0.014 | 0.007 | −2.09 | ** | −0.009 | 0.005 | −1.75 | * | 0.36 | |
| ASC*Income | 0.140 | 0.032 | 4.33 | *** | −0.001 | 0.024 | −0.05 | | 12.20 | *** |
| ASC*Education | −0.009 | 0.027 | −0.32 | | −0.042 | 0.022 | −1.90 | * | 0.87 | |
| ASC*Marital Status | −0.009 | 0.120 | −0.07 | | 0.000 | 0.101 | 0.00 | | 0.00 | |
| ASC*Occupation | −0.761 | 0.164 | −4.65 | *** | 0.151 | 0.110 | 1.38 | | 21.46 | *** |
| ASC*Residence Area | −0.348 | 0.133 | −2.62 | *** | 0.227 | 0.098 | 2.31 | ** | 12.15 | *** |
| *LLF* | −1335.46 | | | | −2138.04 | | | | | |
| Adj. Pseudo $R^2$ | 0.147 | | | | 0.103 | | | | - | |
| No. of Obs. | 1440 | | | | 2178 | | | | | |

Note (1) LR test b/w two clusters: $\chi^2$ (0.01, 15) = 65.62 > 30.58; Note (2) Coefficient comparison w/$\chi^2$ (0.01, 1) = 6.63;$\chi^2$ (0.05, 1) = 3.84; Note (3) ***, **, *: Significance at 1%, 5%, 10% level.

According to the main results, an improvement from the low-level to the medium-level ($\chi^2$ = 4.17, $p < 0.05$), the low-level to high-level ($\chi^2$ = 7.89, $p < 0.01$) of the DM function, and the low-level to high-level ($\chi^2$ = 5.95, $p < 0.05$) of the WM function showed significant differences. Thus, the influence on the choice probability was found to be greater in the high-involvement group than the low-involvement group, although the coefficients had the same directions. In addition, income (t = 4.33) and occupation (t = −4.65) were statistically re-examined as being significant variables only in the high-involvement group. Regarding the local variable residence area, Daejeon region was significantly associated with the high-involvement group (t = −2.62) while the Chungbuk and Chungnam areas correlated to the low-involvement group (t = 2.31). However, there was no significant difference in the effect of age and education variables between the two groups.

Table 6 shows the results of analyzing the marginal MWTPs for the two groups (Regarding the level changes of each attribute for the pooled sample ('Low' to 'Medium' and 'Medium' to 'High'), 4829 KRW and 1848 KRW for DM, KRW 2916 and 1271 KRW for FC, and 6058 KRW and 2663 KRW for WM were derived, respectively). The 95% MWTP confidence intervals were estimated using Krinsky and Robb [75]'s Monte Carlo simulation, and the t-statistics were derived based on the delta method [76]. First, regarding the DM function, the increments in the two levels of the high-involvement group were significant with the result. The different between the low to medium level is KRW 6467 (with a 95% confidence range of KRW 4261 to KRW 8673), and the medium to high level indicates a difference of KRW 2121 (with a 95% confidence range of KRW 382 to KRW 3860). Meanwhile, the low-involvement group disclosed KRW 3924 (from an interval of KRW 2139 to KRW 5693) in terms of the improvement from low to medium level. However, an insignificant effect on the change from medium to high level was detected. Regarding the FC function, there was no statistical difference between the two groups on the change from low to medium level (high involvement at KRW 2783 vs. low involvement at KRW 3036), while the medium level did not effectively move into the high level in both groups.

**Table 6.** Estimates of MWTP by cluster.

| Attribute | Level | Implicit Prices (*t*-Ratio) | Confidence Interval 95% | Implicit Prices (*t*-Ratio) | Confidence Interval 95% |
|---|---|---|---|---|---|
| Drought Management | Low → Mid | 6467 (5.75) *** | [4261–8673] | 3924 (4.33) *** | [2139–5693] |
| | Mid → High | 2121 (2.39) ** | [382–3860] | 1486 (1.81) n.s. | [−114–3098] |
| Flood Control | Low → Mid | 2783 (2.81) *** | [842–4725] | 3036 (3.64) *** | [1407–4672] |
| | Mid → High | 1162 (1.25) n.s. | [−662–2986] | 1228 (6.27) n.s. | [−353–2824] |
| Water-quality Monitoring | Low → Mid | 6471 (5.82) *** | [4290–8652] | 5661 (5.81) *** | [3898–7432] |
| | Mid → High | 3728 (3.89) *** | [1848–5609] | 1951 (2.38) ** | [342–3555] |
| Total MWTP | | 21,570 (10.26) *** | [17,450–25,691] | 14,569 (8.75) *** | [11,306–17,832] |

Note (1) Unit of Marginal WTP: won/year-household. Note (2) ***, **: Significance at 1%, 5% level.

Moreover, the WM function also exhibited significance at each level in both groups as per the results. This indicates that MWTPs (95% confidence interval) of the high-involvement and low-involvement groups, respectively showed KRW 6471(KRW 4290–KRW 8652) and KRW 5661 (KRW 3898–KRW 7, KRW) regarding the change from medium to high level, as well as KRW 3,728 (KRW 1848–KRW 5609) and KRW 1951 (KRW 342–KRW 3555) regarding the change from medium to high level. However, the slope test results showed the intergroup heterogeneity. Finally, the total amount of MWTPs for the high-involvement group was KRW 21,570 (95% confidence interval: [17,450; 25,691]), and that of the low interest group was KRW 14,569 (95% confidence interval: [11,306; 17,832]).Thus, there a merged MWTP difference of around KRW 7000 a year per household between the two groups exists (in US dollar terms, the converted amount is approximately USD 18.58 and USD 12.61, respectively, with a difference of 6.06USD based on the exchange rate system of the Bank of Korea). The large difference in MWTP between high and low climate change awareness groups is in line with Kim et al. [34]'s study, which found that the Daecheong Dam basin was one of the most damaged areas in the summer of 2020, and that damage caused by climate change could worsen in the future.

Both groups showed the same value intensities in the order of water quality monitoring, drought management, and flood control. It is difficult to secure drinking water supplies in neighboring regions, since it continues to suffer from water quality issues such as non-point pollutant sources flowing from surrounding areas of rivers, which has a significant influence on the supply of various water types, such as daily and agricultural water [77]. Based on the findings of these prior research, the greatest MWTP of the survey respondents' water quality monitoring attribute is considered as accurately reflecting the true situation.

The estimated study values indicated results that are different from prior studies [55–58]. Nevertheless, there is agreement on the utility of reducing damages. Granted, the values were limited to tentative results that further research can rectify. These results, however, show that the economic value of the dam's public functions is regulated by climate change awareness. The estimation results can be differentiated from previous studies since the values of the dam's function attributes, corresponding to climate change, can be derived within a single analysis framework and the utility size between the attributes can be compared.

## 4. Conclusions

The purpose of this study was to estimate the economic value of the Daecheong Dam for the public function of responding to climate change. It examined the moderating effect

of climate change perceptions on value estimates by applying choice experiments. The study specified three dam function attributes such as drought management, flood control, and water quality monitoring, and subdivided each into three levels to improve the status quo. Survey data from 603 households living in Daejeon, Chungbuk, and Chungnam were collected to perform the choice experiments. Subsequently, two clusters, including high-involvement and low-involvement groups, were extracted based on the climate-change perception index. According to the main results of comparing the marginal willingness-to-pay between the two clusters, the attributes and price variable significantly affected the choice probability to benefit from improvements in the rational signs of the coefficients. This result does not violate the independence of irrelevant alternatives assumption. The improvement values of high-involvement and low-involvement groups are, respectively, estimated as KRW 21,570 and KRW 14,572 a year per household.

The findings of this study have the following managerial and policy implications. First, the estimates of the economic value of Daecheong Dam for the public function of responding to climate change are the same in both clusters, and were found to be in the order of water quality monitoring, drought management, and flood control. This can be interpreted as the environmental concerns at the study site being fairly reflected, as serious water bloom has occurred in Lake Daecheong since the rainy season in 2016, and because water pollution still needs to be addressed. Moreover, the implementation of a restricted water supply in the Chungnam area in 2015 raised the awareness of national disasters, leading to the perception that drought prevention is a more urgent problem, and demands an immediate countermeasures, in comparison to flood protection. These results show that public projects, for which the levels of public awareness are sufficiently considered, will have higher reception because the related economic value can vary according to public recognition.

As a result, in order to manage future water resources and establish measures to prevent water disasters in consideration of climate change, it is essential to first identify the increasing flood volume and decreasing dry-water volume due to climate change and to establish policies based on citizens' demands. For example, disaster-prevention urban planning, the designation and administration of natural disaster risk improvement zones, and the provision of safe drinking and living water may all boost the policy's positive efficacy.

This study has also demonstrated that the economic value of the dam's public functions are regulated by climate change awareness, which supports the belief that policies in which the public's propensity to climate change is considered can positively promote public welfare. Values of public roles are found for both groups, regardless of climate change perception, but their degree shows remarkable differences between the groups. In particular, the MWTPs on functional improvements are not significant at some levels, suggesting that people may disagree with the use of tax for strengthening its functions; this implies that the actual importance of the dam's role in climate change perception has not been understood fully. Hence, identifying the specific sub-groups according to the awareness level of climate change and building differential communication strategies for a paradigm shift is necessary to heighten the feasibility of implementing dam improvement and development plans. The results of this study can be differentiated from previous ones in that the values of the dam function corresponding to climate change can be derived within a single analysis framework, and the utility value between attributes can be compared. We believe that the obtained economic values translate into suggestions for fulfilling environmental policy needs.

Although it is meaningful, in that this study provides theoretical and practical implications to identify the core priorities of the dam's public functions from the perspective of the value concept, and thus suggest directions of future dam improvement projects, several limitations need to be addressed in further research. First, this study extracts the three attributes as the role of dams for climate change measures through the extensive literature review, expert surveys, and focus group interviews; however, there has been a limit to

the generalizability of these attributes to all cases even though it is considered reasonable to generalize the results of the study when the operating conditions of Daecheong Dam and the demographic characteristics of residents are under similar conditions. Therefore, it is recommended that more common functions appropriate for general cases will be further explored in subsequent studies, and socioeconomic variables reflecting specific regional characteristics will need to be considered more sensitively if the model extension. Second, the estimated value in this study is limited to the tentative results, not conclusive ones, as the choice experiments are based upon the questionnaire survey due to the stated preferences. In fact, since this drawback is an inevitable vulnerability of stated preference experiments, future researchers can develop a methodology to calculate the values of the dam functions using the revealed preference data. Third, at the level setting, the highest level is the expected value for new technology development, but the problem is that we could not explain exactly what kind of technology is anticipated. Regarding this issue, additional research on new technology development will be needed. Fourth, despite the introduction, attributes, and adequate explanation for each level of the study area (Daecheong Dam), the possibility of a Hypothetical Bias cannot be discounted. The need to solve the Hypothetical Bias using various techniques such as cheap talk [78], certainty follow-up [79] and oath [80] is raised. Additionally, a binary discrete choice question is incentive compatible, but multinomial repeated choice is not. In subsequent studies, a research design based on binary discrete choice is required.

Lastly, the conditional logit model requires a strict assumption that it must be accepted by IIA. Accordingly, in this study, the Hausman test was conducted to verify that the IIA assumption was fulfilled, and the conditional logit model was selected as the final research model. Many studies [81–83] stated that IIA assumption is too dependent on the parameterization of the model. The mixed logit model can be considered as an improved alternative that can describe choice probabilities across a given mixing distribution in an adaptable and flexible manner. Various mixing distributions, such as normal, log-normal, triangular, or uniform, can be used, depending on prior information on the taste variation among the decision makers [84]. In subsequent studies, it is necessary to carefully consider these points and select a research model.

**Author Contributions:** Conceptualization, H.O. and H.L.; methodology, H.O. and H.L.; software, H.O. and S.Y.; validation, H.L.; formal analysis, H.O. and S.Y.; investigation, H.O. and S.Y.; resources, H.O. and S.Y.; data curation, H.O. and S.Y.; writing—original draft preparation, H.O. and S.Y.; writing—review and editing, H.O. and S.Y.; visualization, H.O. and S.Y.; supervision, H.L.; project administration, H.L. All authors have read and agreed to the published version of the manuscript.

**Funding:** This research received no external funding.

**Institutional Review Board Statement:** Not applicable.

**Informed Consent Statement:** Informed consent was obtained from all subjects involved in the study.

**Data Availability Statement:** The data will be made available on request from the corresponding author.

**Conflicts of Interest:** The authors declare no conflict of interest.

## Appendix A

Table A1. Factor analysis of climate change perceptions.

| Factors and Items | Loading |
| --- | --- |
| **Level of understanding of climate change** | |
| Understanding the causes (EFA $\alpha$ = 0.776; Eigen value = 2.568; Variance explained = 21.40%) | |
| Recent catastrophic events have been caused by the climate change. | 0.792 |
| Scientific information about the climate change should be provided to every citizen. | 0.776 |
| This is the time to discuss how to adapt to climate change, not to argue. | 0.723 |
| The national counter strategy against the climate change is too passive. | 0.629 |
| Understanding the measures (EFA $\alpha$ = 0.795; Eigen value = 2.555; Variance explained = 21.30%) | |
| Preparation for the climate change must be a primary objective of national policy | 0.852 |
| Recent catastrophic events have been caused by the climate change. | 0.797 |
| Scientific information about the climate change should be provided to every citizen. | 0.776 |
| This is the time to discuss how to adapt to climate change, not to argue. | 0.740 |
| Understanding the results (EFA $\alpha$ = 0.808; Eigen value = 2.363; Variance explained = 19.69%) | |
| Recent catastrophic events have been caused by the climate change. | 0.808 |
| Scientific information about the climate change should be provided to every citizen. | 0.767 |
| This is the time to discuss how to adapt to climate change, not to argue. | 0.692 |
| The national counterstrategy against the climate change is too passive. | 0.621 |
| KMO = 0.860; Bartlett's test of sphericity: $\chi^2$ = 2489.81; df = 66; $p$ = 0.000 | |
| **Level of awareness of the behavioral pattern (EFA $\alpha$ = 0.817; Eigen value = 3.166; Variance explained = 52.76%)** | |
| The environmental protection helps improve the quality of life. | 0.770 |
| With the environmental protection, everybody wins eventually. | 0.765 |
| The health threat of air pollution is more serious than people perceive it to be | 0.721 |
| Global warming is still ongoing. | 0.715 |
| Protecting the environment is beneficial to my health. | 0.694 |
| The climate change is affecting everyone in real time. | 0.690 |
| KMO = 0.820; Bartlett's test of sphericity: $\chi^2$ = 1171.84; df = 28; $p$ = 0.000 | |
| **Level of behavioral style (EFA $\alpha$ = 0.677; Eigen value = 2.363; Variance explained = 39.33%)** | |
| The environmental protection helps improve the quality of life. | 0.784 |
| With the environmental protection, everybody wins eventually. | 0.749 |
| The health threat of air pollution is more serious than people perceive it to be | 0.602 |
| Global warming is still ongoing. | 0.562 |
| Protecting the environment is beneficial to my health. | 0.535 |
| The climate change is affecting everyone in real time. | 0.468 |
| KMO = 0.678; Bartlett's test of sphericity: $\chi^2$ = 664.285; df = 15; $p$ = 0.000 | |

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
