# Peer review of "Willingness to Pay for Public Benefit Functions of Daecheong Dam Operation: Moderating Effects of Climate Change Perceptions"

_sustainability, doi:10.3390/su132414060_

Round 1
Reviewer 1 Report
This is an interesting work that combines various techniques to estimate the economic value of large civil projects for the public benefit function of responding to climate change. this is illustrated with the daecheong dam.
However, the structure of the document needs to be improved.
Line 84: Literature review section.
This section could be taken to the introduction and reoriented in the sense of presenting the problem: how is the economic value of buildings as dams estimated for the public benefit function of responding to climate change?
In "2. Materials and Methods" a subsection is missing briefly describing the study area: the basin where Daecheong Dam is located. It should be clear to the reader, among other things:
* population benefited by the dam
* what does it mean that in section 2.5 it says: "adults aged 20 and older who, directly and indirectly, benefit from the public role of Daecheong Dam"
* Is a sample size N = 630 (210) adequate?
It would also be interesting to briefly describe the specific effects that anthropogenic climate change is causing in the basin.
For readers unfamiliar with the geography of Korea, authors should consider including a figure showing the basin, dam, lake, cities, and all the names used in the paper.
Line 204. N = 630 or 603? missing an explanation?
Lines 292 - 297. Must be in materials and methods
Line 370. Section 4 conclusion
Part of this section is not actually a conclusion, but rather a very good discussion. I recommend that the authors reorganize this section, as a closing of the previous section "3 results and discussion" A very concrete conclusion should be formulated
Author Response
This is an interesting work that combines various techniques to estimate the economic value of large civil projects for the public benefit function of responding to climate change. this is illustrated with the daecheong dam.
However, the structure of the document needs to be improved.
Line 84: Literature review section.
This section could be taken to the introduction and reoriented in the sense of presenting the problem: how is the economic value of buildings as dams estimated for the public benefit function of responding to climate change?
→ I added the contents of previous studies on the public interest function and analysis methodology of the dam. It also added that CE is one of the methods of estimating the economic value of the dam's public interest function. [Line 138]
In "2. Materials and Methods" a subsection is missing briefly describing the study area: the basin where Daecheong Dam is located. It should be clear to the reader, among other things:
* population benefited by the dam
→ Added the 2.1 Study area [Line 108]. In this part, I described Daecheong Dam location, population, water supply rate, and about hydroelectric power plants.
* what does it mean that in section 2.5 it says: "adults aged 20 and older who, directly and indirectly, benefit from the public role of Daecheong Dam"
→ I modified it as “The study population comprised adults aged 20 and older who, directly and indirectly, benefit from the public rolevarious water supply and hydroelectric power generation of Daecheong Dam” [Line 278] Since Korea is obligated to pay taxes for adults over the age of 20, we have selected them as survey subjects who can respond to the payment means of this study.
* Is a sample size N = 630 (210) adequate?
→ Since all survey respondents each have to choose 18 alternatives, the final valid sample will be more. This is not considered to be a small number of samples that are suspected of bias in parameter estimation.
It would also be interesting to briefly describe the specific effects that anthropogenic climate change is causing in the basin.
→ Added the influence of climate change on the river basin. [Line 131]
For readers unfamiliar with the geography of Korea, authors should consider including a figure showing the basin, dam, lake, cities, and all the names used in the paper.
→ Added it as figure 1.[Line 127]
Line 204. N = 630 or 603? missing an explanation?
→ The final 603 valid questionnaires were obtained, excluding the invalid questionnaire due to incorrect, unwritten, and duplicate questionnaires. [Line 288]
Lines 292 - 297. Must be in materials and methods
→ I modified it.[Line 335]
Line 370. Section 4 conclusion
Part of this section is not actually a conclusion, but rather a very good discussion. I recommend that the authors reorganize this section, as a closing of the previous section "3 results and discussion" A very concrete conclusion should be formulated
→ The contents of the comparison between this study and previous studies were moved to 3. Results and diffusion.[Line 475] Along with this, the conclusion part was supplemented(Recommendations, or a policy set for policymakers).[Line 522]

Reviewer 2 Report
1. I recommend that you use simple present tense instead of past present tense in abstract & conclusion
2. The literature review (section 2.1.) is usually a part of the introduction. It should not be in materials and methods.
3. Please move "2.3. Addressing research objectives" to the introduction _ the objectives are discussed in the introduction
4. Please clearly mention the null and research hypothesis.
5. You may want to discuss about the uncertainties of your models and the recommendations for future works under the discussion.
6. I recommend that you remove the citations (prior studies) from the conclusion. You do not have to discuss about other works in the conclusion.
Author Response
- I recommend that you use simple present tense instead of past present tense in abstract & conclusion
→ I modified it.
- The literature review (section 2.1.) is usually a part of the introduction. It should not be in materials and methods.
→ I totally agree with the comment. However, in the case of this study, since it is about the unfamily research target site of Daecheong Dam in Korea, we prepared a literature review focusing on previous studies in Korea. It also supplemented the damage and methodology of the Korean river basin caused by climate change.[Line 131]
- Please move "2.3. Addressing research objectives" to the introduction _ the objectives are discussed in the introduction
→ I modified it. [Line 84]
- Please clearly mention the null and research hypothesis.
→ This study is to derive the willingness to pay for each attribute of the research target site (Daecheong Dam), which has not been studied before. Therefore, I thought there was an exploratory tendency more than confirmatory, so I added a research question instead of the research hypothesis. [Line 98]
- You may want to discuss about the uncertainties of your models and the recommendations for future works under the discussion.
→ I described the limitations of the study due to the Hypothetical Bias and Research model, and added points to be improved in subsequent studies.[Line 565]
- I recommend that you remove the citations (prior studies) from the conclusion. You do not have to discuss about other works in the conclusion.
→ The comparison of conclusions with previous studies was revised to 3. Results and Discussion.[Line 475] We also supplemented the discussion from previous studies.[Line 464]
Reviewer 3 Report
This case study investigates the economic value of Daecheong Dam for the public function and the interrelations with climate change perception by conducting surveys to perform choice experiments. Taking public perception into account is substantially important in infrastructure projects, and thus this study has the potential to contribute to the literature after addressing my following comments.
- The manuscript requires extensive English editing and proofreading.
- I could not find any robust discussion in “Results and discussion” section. I expect to see confirming and contrasting studies with your results. This section only presents the results.
- There should be recommendations, or a policy set, for policymakers or decision makers to mitigate adverse effects and improve the positive impacts.
Author Response
This case study investigates the economic value of Daecheong Dam for the public function and the interrelations with climate change perception by conducting surveys to perform choice experiments. Taking public perception into account is substantially important in infrastructure projects, and thus this study has the potential to contribute to the literature after addressing my following comments.
- The manuscript requires extensive English editing and proofreading.
→ We will improve the completeness of the thesis through English editing and proreading.
- I could not find any robust discussion in “Results and discussion” section. I expect to see confirming and contrasting studies with your results. This section only presents the results.
→ The contents were supplemented by investigating previous studies supporting the results of this study. [Line 464] I described the results of this study through comparison with previous studies.[Line 475]
- There should be recommendations, or a policy set, for policymakers or decision makers to mitigate adverse effects and improve the positive impacts.
→ Policy implementation based on consumer needs is required, and detailed policy examples have been added in the conclusion.[Line 522]
Reviewer 4 Report
See attached

Author Response
Major
- DCE Design and Description. You mention that you described the 3 levels for DM, FC, and WM as status quo partial and substantial improvement, but more detail is necessary. Most importantly, you did not describe the increased level of protection afforded by the low, medium, and high levels. For example, if the probability of a flood disaster is 15% for status quo for FM, then perhaps it is 10% for medium level, and 5% for high level. Without being more specific with the outcomes, you do not know what your attribute levels are measuring such that your WTP outcomes only represent ordinal preferences. Without being tied to specific information, your WTP outcomes cannot be tied to specific benefit-cost analysis. What technology did you describe? This would explain your outcomes in Table 6, specifically why High is not perceived as better than Medium. Without a better description, you may be observing WTP for medium because people want to know they're making a contribution (i.e. warm glow (Andreoni 1989)) without thinking about the actual level of quantity/service being provided. There is no way to resolve this major shortcoming without collecting new data.
→ The survey respondents of this study have various demographic characteristics and filled out the questionnaire in general and intuitive words to minimize the bias according to the respondents' understanding. And in the original questionnaire, we prepared the criteria for the survey respondents through a brief description of DM, FC, and WM. Nevertheless, I agree with the opinion that the explanation of “High level” is somewhat abstract, and I clearly describe it as a limitation of the study in the conclusion chapter.[Line 567]
Your opt-out alternative seems to have been communicated incorrectly. "Choose neither option" may seem to imply to the person that they are opting out of making a purchase. In reality, each person is obligated to continue to pay for their current service level. The opt-out is a decision to stay at the status quo and associated expense, not to opt-out of services entirely.
→ All questionnaires consist of higher-level alternatives than status quo for at least one of the DM, FC, and WM attributes. The responses that did not select any alternatives(i.e. "choosing neither option") were considered to have chosen to remain at the current level. This content has been added to the 2.4 Development of Measurement instrument.[Line 273]
Do participants usually pay for their water on an annual basis? In many water studies, it most common to see monthly billing, in order to facilitate comprehension and easier response by the survey respondent who is already used to thinking about monthly billing. Why did you use annual?
→ Korea's water bill is usually paid monthly. However, in the case of this study, WTP was calculated in the form of a voluntary contribution called Financial Support, as expressed in the original questionnaire. In a study by Chipfupa, U., & Wale, E. (2019), the payment period of WTP for water bills was set on a yearly basis. In a Korean study, WTP for water quality improvement was investigated on an annual basis in a study by Jae Bok Lee, Yoon Kyoung Cho, and Sang In Park (2012), and in a study by Jul Hyun Jeon, Chung Sun Lee, and Hi Jun Shin (2010), WTP for water quality improvement was also investigated as annual payments.
To clarify the term, existing “water use charge” is changed to “financial support”, which is the term used in the questionnaire.
- DCE format/Incentive compatibility. When eliciting a public good, everyone potentially benefits from its implementation and pays for the good, except free-riders. This is why it should instead be posed as a referendum DCE (vote yes or vote no), not as multinomial choice. Under certain conditions, binary repeated choice DCE are incentive compatible, but multinomial repeated choice are not. Please see Carson, Groves, and List (2014) and Vossler, Doyon, and Rondeau (2012) for additional insight. There is no way to resolve this major shortcoming without collecting new data.
→ As in the comment, I agree with the opinion that the binary discrete choice question may have incentive compatible, but the multinomial choice method does not. This is described as a limitation of the study.[Line 567]
- DCE modeling: Running a conditional logit is an inferior model when modeling preferences for discrete choice data. At minimum, I would expect to see the mixed/random parameters logit model, which is almost ubiquitously shown as the base model across the DCE literature because it almost always has a better model fit than a conditional logit model.
The justification of the conditional logit model presupposes the assumption of Independence from Irrelevant Alternatives. Through the analysis, it was confirmed that the IIA assumption was fulfilled (Table 3.), and the conditional logit model was adopted as the research model accordingly. According to Dahlberg, M., & Eklöf, M. (2003), the mixed logit (MXL) model, and the multinomial probit (MNP) model estimators lead to exactly the same conclusions as the traditional conditional logit (CL) model. Nevertheless, studies by Fry, T. R., & Harris, M. N. (1996; 1998), Cheng, S., & Long, and J. S. (2007) suggest that IIA assumption is overly dependent on the parameterization of the model. It is added as a limitation of the study.[Line 570]
- Hypothetical Bias remains a major issue in stated preference studies such as yours with several recent papers demonstrating as much (Penn and Hu 2018). Typically, hypothetical bias is when stated WTP is higher than real WTP. Multiple tactics exist to mitigate this form of bias such as cheap talk (Penn and Hu 2019), certainty follow-up (Blomquist, Blumenschein, and Johannesson 2009), oath, among others. Unless you simply did not mention this in the submitted version, there is no way to resolve this major shortcoming without collecting new data.
→ Despite the introduction, attributes, and adequate explanation for each level of the study area (Daecheong Dam), the possibility of Hypothetical Bias cannot be excluded. I described in the conclusion part so that the research can be developed through these pre- and post-measures.[Line 564]
- Table 5: Why did you decide to do interactions of the "Choose neither" ASC? Why not run these interactions with the other attributes? It would be interesting to see whether DM, FC, or WM is relatively more important to certain groups.
→ Alternative specific constant (ASC) affects the selection of the choice set, but represents the average of all influences not included in the model. In order to understand how respondents' socio-demographic variables affect the probability of selection, the interaction terms between the socio-demographic variables and ASC were additionally included in the model. Since the values of socioeconomic variables such as income or age do not vary among the alternatives, it is impossible to estimate the model when including them.
Therefore, all variables (socio-demographic variables) that do not differ between alternatives can be estimated by creating an interaction term multiplied by variables (i.e. ASC) that differ between alternatives.
- Table 5/6: Rather than test whether there is a significant difference in the WTP between medium and high, you should instead test whether a significant difference in preferences exists in the coefficients of Table
→ One of the ways to compare the two research models is the Likelihood Ratio Test (Marques, F.J., Coelho, C.A. & Rodriguez, P.C. (2017); J. Park, B. Sinha, A, Shah, D, Xu and Lin (2015). As a result of the likelihood ratio test, it was found that there was a significant difference between the climate change awareness high-involvement group and the low-involvement group analysis model.
In addition, as a result of comparing the coefficient values between groups through the slope test, there was a significant difference in the DM-mid, DM-high, and WM-hign attributes, and all of them showed a high probability of selection in the high group.
- Table 6: The typical approach is to show the WTP from the base level (in this case "low") to each level of the attribute, not between levels. I say this because your results are misleading. WTP is still statistically significant compared to the reference level (i.e. status quo).
→ The high level of this study assumes that new technology development is applied for each attribute (DM, FC, and WM). Therefore, we decided that it would be more meaningful to derive a difference between levels because MWTP for the alternative from Middle level to High level means the willingness to pay for the introduction of new technologies.
- Wording: I strongly encourage the authors to reread the paper to be more plain spoken. At numerous points, I did not understand or thought the points being made were misdescribed, so much so that it was a distraction. Please have the manuscript reviewed again to clarify ideas.Several examples appear below:
- L44-45: "more than economic ones". What does this mean? I understand that several aspects of dams are public goods (electricity is excludable, so not public), but not this comparison. Do you mean dams are not a private good?
→ In the case of water resources, they used to be uniformly classified as public goods, but as the water shortage intensifies and the use of water diversifies, they have some characteristics of private goods. Water resources can be said to strengthen the market goods of water as the population increases and industries advance under limited conditions.
In addition, since there are areas that directly benefit from the dam (part of Korea, including Daejeon Metropolitan City in this study) and have the characteristics of exclusion, it also means that it has the characteristics of private goods, not complete public goods.
To clarify, it was modified to "dams is characterized by public goods more than private goods".[Line 46]
- Highlights: What does "lower age and higher income levels have more choice possibilities for improvement alternatives" mean?
→ As in question 5, an ASC interaction term was created to estimate the probability of selection of demographic variables, and age was significant in the negative (-) direction and income in the positive (+) direction.
This means that there is a higher probability of selecting an improved alternative from at least one attribute than the current level. This is the result of interpreting the direction of the sign.
- L75: Avoid "was selected for problem diagnosis" and be more plain spoken such as: "This study investigates the benefits of Daecheong Dam, located in Daejeon metropolitan city, South Korea."
→ I have Modified it [Line 76]
Minor
- Table 2 and L204: To clarify, by "(210 copies for each type)" and "Type A/B/C" in Table 2, you mean per block of the DCE, correct? Type was not previously used.
→ I have modified it. [Line 257]
- Table 2: The convention is to say 20-29 rather than "20s".
→ I have modified it.
- Education. Your levels are either incomplete or not descriptive enough. I can't tell whether a person who completed some, but did not finish high school is still included in the "High school" category.
→ In the original questionnaire, checking the number of years of education according to the Korean education system was required.(6 years in elementary school, 3 years in middle school, 3 years in high school, etc.). So "high school" means in here is high school enrollment and high school graduation.
- The occupation variable is more elaborate than is standard. Typical categories include Full-time, Part time, student, unemployed, retired, homemaker/housewife. Area of profession (clerical, production, service) are atypical.
→ We classified in detail to reflect the demand (alternative selection and willingness to pay) for various occupations of the survey respondents.
- HPM: You only uses the acronym twice, so recommend removing to reduce the number of acronyms in the paper.
→ Removed the acronym and modified it to the hedonic price method.
- Grammatical/English/Spelling
- o L37: "Highly required": odd wording
→ I have modified it.[Line 38]
- o L62: Benefitting: spelling
→ I have modified it.[Line 63]
- o "lowly involved"
→ I have modified it.[Table 5]
<Reference>
- Chipfupa, U., & Wale, E. (2019). Smallholder willingness to pay and preferences in the way irrigation water should be managed: a choice experiment application in KwaZulu-Natal, South Africa. Water SA, 45(3 July). (https://doi.org/10.17159/wsa/2019.v45.i3.6735)
- Jae Bok Lee, Yoon Kyoung Cho, Sang In Park. (2012). Estimating the Economic Benefits of Improving Tap Water Quality Using a Discrete Choice Model. Korean Journal of Public Administration, 50(3), 327-351.
- Chul Hyun Jeon, Chung Sun Lee, Hio Jung Shin. (2010). Estimation of Welfare Change from Water Quality Degradation. Journal of environmental policy, 9(2), 135-155.
- Dahlberg, M., & Eklöf, M. (2003). Relaxing the IIA assumption in locational choice models: A comparison between conditional logit, mixed logit, and multinomial probit models. Nationalekonomiska institutionen.
- Fry, T. R., & Harris, M. N. (1996). A Monte Carlo study of tests for the independence of irrelevant alternatives property. Transportation Research Part B: Methodological, 30(1), 19-30.( https://doi.org/10.1016/0191-2615(95)00019-4)
- Fry, T. R., & Harris, M. N. (1998). Testing for independence of irrelevant alternatives: some empirical results. Sociological Methods & Research, 26(3), 401-423. (https://doi.org/10.1177/0049124198026003005)
- Cheng, S., & Long, J. S. (2007). Testing for IIA in the multinomial logit model. Sociological methods & research, 35(4), 583-600.( https://doi.org/10.1177/0049124106292361)
- Marques, F.J., Coelho, C.A. & Rodrigues, P.C.(2017) Testing the equality of several linear regression models. Comput Stat 32, 1453–1480. (https://doi.org/10.1007/s00180-016-0703-1)
- Park, B. Sinha, A, Shah, D, Xu, and J. Lin(2015) Likelihood Ratio Tests for Interval Hypotheses with Applications, Commun. Stat.,44, 11, 2351-2370 (https://doi.org/10.1080/03610926.2013.781639).

Round 2
Reviewer 1 Report
The manuscript improved with respect to the previous version. The points that worried me have been solved. The writing is difficult, a style check and grammar correction might be a good idea.
Author Response
The manuscript improved with respect to the previous version. The points that worried me have been solved. The writing is difficult, a style check and grammar correction might be a good idea.
- In order to deliver a clear message and improve the overall completeness of the paper, we will implement Professional English proofreading.
Reviewer 3 Report
The authors mostly addressed my concerns. However, the manuscript still requires English proofreading, it has typos (for instance, "... the beneficts of..." in line 75) and language problems. There are some font problems in new text added in the revision. Apart from these, this manuscript has a potential to contribute the related literature.
Author Response
The authors mostly addressed my concerns. However, the manuscript still requires English proofreading, it has typos (for instance, "... the beneficts of..." in line 75) and language problems. There are some font problems in new text added in the revision. Apart from these, this manuscript has a potential to contribute the related literature.
- In order to deliver a clear message and improve the overall completeness of the paper, we will implement Professional English proofreading.
Reviewer 4 Report
Multiple grammatical and typographical errors are still present including "beneficts" (L75), "cheaptalk" (L546), and "lowly involved group" (Table 5)
Eliminate discussion of CL vs MXL vs MNP. It is misplaced. Dahlberg and Eklof is a very old and not well known citation versus, for example, Fiebig et al (Marketing Science, 2010), Ken Train, David Hensher, or other well known discrete choice econometricians in the field. Further, an easy way to verify that MXL is better is to run the model and then examine goodness of fit, which is virtually guaranteed to be an improvement over CL based on the numerous models I have run as well as papers I have read and reviewed. Better models would relax several of the restrictions from CL including IIA, uniform preferences (MXL), uniform scale heterogeneity (Scale logit), or both (GMNL).
Author Response
Multiple grammatical and typographical errors are still present including "beneficts" (L75), "cheaptalk" (L546), and "lowly involved group" (Table 5)
- I have modified it.
- In order to deliver a clear message and improve the overall completeness of the paper, we will implement Professional English proofreading.
Eliminate discussion of CL vs MXL vs MNP. It is misplaced. Dahlberg and Eklof is a very old and not well known citation versus, for example, Fiebig et al (Marketing Science, 2010), Ken Train, David Hensher, or other well known discrete choice econometricians in the field. Further, an easy way to verify that MXL is better is to run the model and then examine goodness of fit, which is virtually guaranteed to be an improvement over CL based on the numerous models I have run as well as papers I have read and reviewed. Better models would relax several of the restrictions from CL including IIA, uniform preferences (MXL), uniform scale heterogeneity (Scale logit), or both (GMNL).
- I have cited a study that the mixed logit model is more adaptable and flexible than conditional model because it can be used various mixed distribution(e.g. normal, log-normal, triangular, uniform).
Reference
- Shi, H., & Yin, G. (2018). Boosting conditional logit model. Journal of choice modelling, 26, 48-63.